# Thiodiketopiperazines Produced by *Penicillium crustosum* and Their Activities to Promote Gastrointestinal Motility

**DOI:** 10.3390/molecules24020299

**Published:** 2019-01-15

**Authors:** Xin He, Jing Yang, Ling Qiu, Dan Feng, Feng Ju, Lu Tan, Yu-Zhi Li, Yu-Cheng Gu, Zhen Zhang, Da-Le Guo, Yun Deng

**Affiliations:** 1The Ministry of Education Key Laboratory of Standardization of Chinese Herbal Medicine, State Key Laboratory, Breeding Base of Systematic Research Development and Utilization of Chinese Medicine Resources, School of Pharmacy, Chengdu University of Traditional Chinese Medicine, Chengdu 611137, China; 18380456815@163.com (X.H.); qiuling0308@gmail.com (L.Q.); 18408210828@163.com (D.F.); jufeng18482114870@163.com (F.J.); tltanlu@163.com (L.T.); liyuzhi5654@163.com (Y.-Z.L.); zhangzhendr@126.com (Z.Z.); 2State Key Laboratory of Phytochemistry and Plant Resources in West China, Kunming Institute of Botany, Chinese Academy of Sciences, Kunming 650201, China; yangjingc@mail.kib.ac.cn; 3Syngenta Jealott’s Hill International Research Centre, Berkshire RG42 6EY, UK; yucheng.gu@syngenta.com

**Keywords:** *Penicillium crustosum*, *Isaria cicadae*, thiodiketopiperazines, Mosher analysis, ECD calculation, gastrointestinal motility

## Abstract

Three new thiodiketopiperazines (**1**–**3**), along with two known analogues (**4** and **5**), were isolated from the fermentation broth of *Penicillium crustosum*. Their structures were elucidated through extensive spectroscopic analysis and the absolute configurations of new compounds were determined by Mosher ester analysis and calculated ECD spectra. Compound **4** and **5** have the activity to promote the gastrointestinal motility of zebrafish via acting on the cholinergic nervous system.

## 1. Introduction

Natural products have played an important role in the discovery of medicinal agents and fungi are valuable resources for compounds with bioactivity [1]. *Penicillium crustosum* is a microscopic fungus of the genus *Penicillium*, which is a promising source of bioactive compounds with diverse structures [2]. A wide range of secondary metabolites have been isolated and identified from *P. crustosum*, including alkaloids [3,4], diketopiperazines [5], and polyketides [6,7]. 

Gastrointestinal intolerance is a common preclinical finding, and can be a serious safety concern in the clinic [8]. Due to the gastrointestinal similarity between zebrafish and mammalian [9], zebrafish has recently been used to study functional disorders of the human intestine (inflammatory bowel disease) using methods derived from mammalian models [10]. In the course of our ongoing research on secondary metabolites with activities to promote gastrointestinal motility from terrestrial fungi derived from traditional Chinese medicines (TCM), *P. crustosum*, an *Isaria cicadae*-colonizing fungus, was obtained. After large scale fermentation (36 L), extraction, and column chromatography separation, one fraction showed the obvious activity of promoting gastrointestinal motility on zebrafish. Three novel thiodiketopiperazines (**1**–**3**) from the bioactive fraction, along with two known analogues (**4**–**5**), were isolated and identified (Figure 1). In this paper, the fermentation, isolation, structural elucidation, and bioactivity of compounds **1**–**5** were described.

## 2. Results and Discussion

Compound **1** was obtained as a yellow gum, HRESIMS ion at *m/z* 456.1502 [M + Na]^+^ revealed its molecular formula to be C_20_H_30_N_2_O_5_S_2_ (calculated for C_20_H_30_N_2_O_5_S_2_Na^+^, 456.1488). The IR spectrum of **1** showed the presence of hydroxyl (3420.6 cm^−1^), methylene (2927.7 cm^−1^), and olefinic (1623.0 cm^−1^) groups. The ^1^H NMR spectrum displayed a pair of *p*-disubstituted phenyl proton signals at *δ* 6.85 (2H, d, *J* = 8.7 Hz, H-10 and H-12) and 7.11 (2H, d, *J* = 8.7 Hz, H-9 and H-13); two signals of an oxymethylene at *δ* 4.23 (1H, dd, *J* = 10.0, 3.0 Hz, H-1′a) and *δ* 3.89 (1H, dd, *J* = 10.0, 7.7 Hz, H-1′b); a oxymethine signal at *δ* 3.72 (1H, td, *J* = 4.5, 2.2 Hz, H-2′); two signals of a methylene at *δ* 3.58 (1H, d, *J* = 14.0 Hz, H-7a) and *δ* 3.14 (1H, d, *J* = 14.0 Hz, H-7b); and six methyl signals at *δ* 3.20 (3H, s, 4-Me), 3.01 (3H, s, 1-Me), 1.95 (3H, s, 2-SMe), 1.58 (3H, s, 5-SMe), 1.23 (3H, s, 4′-Me), and 1.21 (3H, s, 5′-Me). The ^13^C NMR and HSQC spectra of compound **1** exhibited 18 carbon signals, including two amides at *δ* 165.4 (C-3) and 165.2 (C-6); four olefinic carbons at *δ* 159.6 (C-11), 132.5 (C-9 and C-13), 115.8 (C-10 and C-12), 128.1 (C-8); one oxymethine carbon signal at *δ* 77.1 (C-2′); one oxymethylene carbon signal at *δ* 70.7 (C-1′); one methylene carbon at *δ* 41.0 (C-7); six methyl signals at *δ* 33.2 (4-Me), 30.6 (1-Me), 26.7 (C-4′), 25.6 (C-5′), 13.7 (5-SMe), and 12.6 (2-SMe); and three quaternary carbons at *δ* 76.9 (C-2), 71.9 (C-3′) and 66.0 (C-5). The HMBC correlations of H-4′, H-5′/C-3′; H-4′, H-5′/C-2′, as well as the ^1^H–^1^H COSY correlations of H-2′/H-1′, indicated the presence of a dimethyltriol moiety; the HMBC correlations of 1-Me/C-2, C-6; 4-Me/C-3, C-5; H-5/C-6; 2-SMe/ C-2, and 5-SMe/C-5 indicated the presence of a thiodiketopiperazines (Figure 2). The primary structure of compound **1** was finally confirmed by the HMBC correlations of H-1′/C-11, H-7/C-8, and H-7/C-3.

The NOESY correlations between H-5 (*δ* 4.92) and 2-SMe (*δ* 1.95) indicated they were on the same side of the diketopiperazine core (Figure 3), and compound **1** should be an epimer of bilain B [11] (Figure 1). The absolute stereochemistry of the triol chain was determined as *S* by modified Mosher’s method, and the absolute configuration of the diketopiperazine core was elucidated as 2*S*,5*R* by quantum chemical ECD calculations, as shown in Figure 4.

Compound **2** was isolated as a yellow gum. Its molecular formula was determined as C_21_H_32_N_2_O_5_S_2_ by the HRESIMS peak at 479.1665 (calcd 479.1645). The NMR data was similar to that of **1**, and differences between **2** and **1** were a methoxyl signal (δ_H_ 3.26; δ_C_ 49.4) instead of a hydroxyl at C-3′, which created an oxymethyl signal and a HMBC correlation between 3′-OMe and C-3′. The NOESY correlations between 2-SMe (δ 2.16) and 5-SMe (δ 2.29) that were observed indicated that these two thiomethyls should be in the same side of the diketopiperazine core (Figure 3). Subsequently, modified Mosher’s method and experimental ECD analysis were applied to confirm the absolute configuration of the C-2′. The absolute configuration of **2** was finally elucidated as 2S,5S,2′S.

Compound **3** shares the same molecular formula with **2** (C_21_H_32_N_2_O_5_S_2_), which was deduced from the HRESIMS spectrum. On comparing its one-dimensional and two-dimensional NMR spectra with that of compound **2**, it appeared that they might be a pair of epimers. The NOESY correlations between H-5 (δ 4.59) and 2-SMe (δ 1.95) were observed, and indicated they were on the same side of the diketopiperazine ring. The experimental ECD spectrum of **3** was similar to those of **1**, and the absolute configuration of C-2′ also was confirmed by modified Mosher’s method analysis. Thus, the absolute configuration of compound **3** should be 2S,5R,2′S.

The structures of compounds **4** [12] and **5** [5] were elucidated on the basis of the NMR and MS spectroscopic comparison with those reported in the literatures.

Compounds **1**–**5** were evaluated for their activity of promoting the gastrointestinal motility of zebrafish treated with Nile red. Compound **4**–**5** significantly promoted the Nile red excretion at doses >8 μM and >16 μM, compared with positive control neostigmine at doses >2 μM. Compound **1**–**3** were inactive, which indicated that the isopentene group might be the pharmacophore.

As we know, acetylcholine is a cholinergic neurotransmitter that binds to acetylcholine receptors selectively, and atropine is a tropane alkaloid muscarinic antagonist which can reduce the excitability of gastrointestinal smooth muscle, as well as reduce the amplitude and frequency of peristalsis by inhibiting the action of acetylcholine [8]. To investigate the underlying mechanism of the gastrointestinal motility-promoting effects, we examined the interfering effect of atropine on the gastrointestinal activation of compound **4**. As shown in Figure 5b,c, at the concentration of 40 μM, compound **4** promoted the gastrointestinal motility significantly and, at 2 μM and 9 μM, atropine decreased the promoting activation of **4** in a dose dependent manner. These results indicated that compound **4** and **5** might promote the gastrointestinal motility of zebrafish via acting on the cholinergic nervous system.

## 3. Materials and Methods

### 3.1. General Experimental Procedures

Optical rotations were determined on a Perkin-Elmer-241 polarimeter (Perkin Elmer, Inc., Waltham, MA, USA) at room temperature. UV spectra were recorded on a Perkin-Elmer Lambda 35 UV–VIS spectrophotometer (Perkin Elmer, Inc., Waltham, MA, USA). IR spectra were measured by Perkin-Elmer one FT-IR spectrometer (KBr) (Perkin Elmer, Inc., Waltham, MA, USA). CD spectra were recorded on a Chirascan circular dichroism spectrometer (Applied Photophysics Ltd., Leatherhead, UK) 1D and 2D NMR were carried out on a Bruker-Ascend-400 MHz instrument (Bruker, Bremen, Germany) at 300 K, with TMS as internal standard. HRESIMS were measured using a Synapt G2 HDMS instrument (Waters Corporation Milford, MA, USA). Preparative HPLC was performed on a NP7000 serials pump (Hanbon Sci. & Tech., Jiangsu, China) equipped with a Kromasil RP-C_18_ column (10 × 250 mm, 5 μm, Akzo Nobel Pulp and Performance Chemicals AB, Bohus, Sweden) using a NU3000 serials UV detector (Hanbon Sci. & Tech., Jiangsu, China). Column chromatography (CC) was performed with silica gel (200-300 mesh, Qingdao Haiyang Chemical Co., Qingdao, China) and Sephadex LH-20 (GE-Healthcare Bio-Sciences AB, Uppsala, Sweden). Zebrafish breeding system (Beijing Aisheng Technology Development Co., Ltd.). Zebrafish were incubated in MGC-100 constant temperature incubator (Shanghai Yiheng Scientific Instrument Co., Ltd., Shanghai, China). Fluorescence intensity of the zebrafish gastrointestinal tract was observed by M165-FC type fluorescence microscopy imaging system (Leica, Wetzlar, Germany). Sodium carboxymethyl cellulose (CMC-Na, batch number: M0202A) and Nile red (lot number: D1219A) were provided by Dalian Meilun Biotechnology Co., Ltd ( Dalian, Chian). All the solvent used were of analytical grade.

### 3.2. Fungal Material and Culture

*P. crustosum* was isolated from the fruiting body of *Isaria cicadae*, which was collected from the suburb of Ya’an, Sichuan province, China, and identified by morphological observation and sequence analyses of the ITS region of rDNA. The strain (GenBank accession No. MK285663) was deposited at Chengdu University of TCM. This fungus was cultivated on a 36 L scale using 1 L Erlenmeyer flasks containing 400 mL of the seed PDA liquid medium for 7 days, and fermentation medium (soluble starch 80 g/L, peptone 5 g/L, NaCl 2 g/L, CaCO_3_ 2 g/L, MgSO_4_∙7 H_2_O 0.5 g/L, K_2_HPO_4_ 0.5 g/L) for 14 days at 28 °C on a rotary shaker (250 rpm).

### 3.3. Fractionation and Isolation

The fermentation broth (36 L) of *P. crustosum* was filtered. The filtrate was extracted with petroleum and followed by EtOAc. The EtOAc solution was dried under vacuum and yielded 14.2 g extract. The extract was separated into 11 fractions by CC on silica gel (300–400 mesh), eluting stepwise with a petroleum ether/acetone gradient (100:0; 98:2; 96:4; 94:6; 92:8; 90:10; 85:15; 80:20; 70:30; 50:50; 0:100). The fractions between eluted with 96:4 and 94:6 were combined (1.35g) and separated further by subjecting to Sephadex LH-20 column chromatography (4 × 180 cm; CHCl3/MeOH, 1:1) and afforded 4 subfractions (Fr.1–Fr.4). Fr.2 (0.42g) was purified by a preparative HPLC using a reversed-phase column to afford **1** (22.1 mg, MeOH–H_2_O, 60:40, 4 mL/min, t_R_: 9.4 min), **2** (5.2 mg, MeOH–H_2_O, 60:40, 4 mL/min, t_R_: 22.3 min), **3** (2.8 mg, MeOH–H_2_O, 60:40, 4 mL/min, t_R_: 15.8 min), **4** (5.8 mg, MeOH–H_2_O, 70:30, 4 mL/min, t_R_: 29.1 min) and **5** (1.8 mg, MeOH–H_2_O, 70:30, 4 mL/min, t_R_: 22.8 min).

Compound **1**. Yellow gum, ${\left[\alpha \right]}_{D}^{20}$ = +9.53 (c = 0.01, MeOH), IR (KBr): 3420.6, 2927.7, 1623.0, 1384.7, 1094.8 cm^−1^; UV(MeOH) λ_max_: 207.1 (4.17), 225.8 (4.03), 275.0 (3.10); ECD (c 6.80 × 10^−4^ M, MeOH) Δε_200 nm_ + 2.16, Δε_209 nm_ + 2.40, Δε_223 nm_ − 4.38, Δε_233 nm_ − 3.25, Δε_242 nm_ −3.51, Δε_268 nm_ − 2.04, Δε_277 nm_ − 2.20; HRESIMS *m/z* 465.1502 [M + Na]^+^ (calcd 465.1488); ^1^H NMR and ^13^C NMR data, see Table 1.

Compound **2**. Yellow gum, ${\left[\alpha \right]}_{D}^{20}$ = −27.88 (c = 0.01, MeOH), IR (KBr): 3524.8, 2926.2, 1623.1, 1384.6, 1087.8 cm^−1^; UV(MeOH) λ_max_: 206.9 (4.19), 226.0 (4.07), 275.1 (3.14); ECD (c 2.10 × 10^−3^ M, MeOH) Δε_194 nm_ + 34.33, Δε_218 nm_ − 12.94, Δε_233 nm_ − 0.63; HRESIMS *m/z* 479.1665 [M + Na]^+^ (calcd 479.1645); ^1^H NMR and ^13^C NMR data, see Table 1.

Compound **3**. Yellow gum, ${\left[\alpha \right]}_{D}^{20}$ = +13.25 (c = 0.01, MeOH), IR (KBr): 3422.8, 2947.4, 1644.2, 1521.5, 1385.4, 1246.0 cm^−1^; UV(MeOH) λ_max_: 206.7 (4.18), 225.8 (4.06), 275.3 (3.13); ECD (c 1.40 × 10^−3^ M, MeOH) Δε_200 nm_ + 8.71, Δε_210 nm_ + 5.37, Δε_224 nm_ − 11.24, Δε_268 nm_ − 5.15,Δε_283 nm_ − 4.27, Δε_285 nm_ − 4.01; HRESIMS *m/z* 479.1653 [M + Na]^+^ (calcd 479.1645); ^1^H NMR and ^13^C NMR data, see Table 1.

### 3.4. Preparation of the (S)- and (R)-MTPA Ester Derivatives of ***1−3***

Compounds **1**–**3** (1.0 mg) were transferred into clean NMR tubes, and were dried completely under the vacuum of an oil pump. Deuterated pyridine (0.35 mL) and (*R*)-(−)-α-methoxy-α- (trifluoromethyl) phenylacetyl chloride (25 µL) were added into the NMR tube immediately, and then the NMR tube was shaken carefully to mix the sample and MTPA chloride evenly. The reaction NMR tubes were permitted to stand overnight at room temperature to afford the (*S*)-MTPA esters of the secondary hydroxyl group. Selected ^1^H NMR data of the (*S*)-MTPA ester derivatives (2′*S*) of **1**–**3** (400 MHz, pyridine-*d*_5_; data were obtained from the reaction NMR tubes directly and were assigned ^1^H, HSQC spectra). In the manner described for 2′*S* above, another portion of compound **1**–**3** (1.0 mg) were reacted in NMR tubes with (*S*)-(+)-α-methoxy-α-(trifluoromethyl)phenylacetyl chloride (25 µL) at room temperature overnight, using deuterated pyridine (0.35 mL) as solvent, to afford the (*R*)-MTPA derivatives of **1**–**3**.

^1^H NMR data (400 MHz, C_5_D_5_N) for (*S*-MTPA) ester of **1**: *δ*_H_ 4.67 (1H, dd, *J* = 10.7, 2.1 Hz H-1′a), 4.29 (2H, dd, *J* = 10.7, 9.0 Hz, H-1′b), 1.51 (3H, s, H-4′), 1.51 (3H, s, H-5′).^1^H NMR data (400 MHz, C_5_D_5_N) for (*R*-MTPA) ester of **1**: *δ*_H_ 4.82 (1H, dd *J* = 10.8, 2.0 Hz, H-1′a), 4.47 (2H, dd, *J* = 10.7, 9.1 Hz, H-1′b), 1.44 (3H, s, H-4′), 1.40 (3H, s, H-5′).^1^H NMR data (400 MHz, C_5_D_5_N) for (*S*-MTPA) ester of **2**: *δ*_H_ 4.37 (1H, dd, *J* = 10.6, 2.2 Hz H-1′a), 4.12 (2H, dd, *J* = 10.7, 9.0 Hz, H-1′b), 1.28 (3H, s, H-4′), 1.26 (3H, s, H-5′).^1^H NMR data (400 MHz, C_5_D_5_N) for (*R*-MTPA) ester of **2**: *δ*_H_ 4.48 (1H, dd *J* = 10.7, 2.1 Hz, H-1′a), 4.27 (2H, dd, *J* = 10.7, 9.1 Hz, H-1′b), 1.19 (3H, s, H-4′), 1.17 (3H, s, H-5′).^1^H NMR data (400 MHz, C_5_D_5_N) for (*S*-MTPA) ester of **3**: *δ*_H_ 4.41 (1H, dd, *J* = 10.7, 2.1 Hz H-1′a), 4.15 (2H, dd, J = 10.7, 9.0 Hz, H-1′b), 1.26 (3H, s, H-4′), 1.24 (3H, s, H-5′).^1^H NMR data (400 MHz, C_5_D_5_N) for (*R*-MTPA) ester of **3**: *δ*_H_ 4.46 (1H, dd *J* = 10.7, 2.0 Hz, H-1′a), 4.25 (2H, dd, *J* = 10.7, 9.0 Hz, H-1′b), 1.17 (3H, s, H-4′), 1.16 (3H, s, H-5′).

### 3.5. Computational Details

The theoretical calculations of compounds **1** and **2** were performed using Gaussian 09 [13]. Conformational analysis was initially carried out using Accelrys Discovery Studio 2.5 to generate conformations by Boltzmann Jump, then minimized by Smart Minimizer using the MMFF molecular mechanics force field. All geometries with relative energy from 0–5.0 kcal/mol were used in optimizations at the B3LYP/6-31G(d). Room temperature equilibrium populations were calculated according to the Boltzmann distribution law (Figure 6). The theoretical calculation of ECD was performed using TDDFT at the B3LYP/6-31G (d, p) level in the methanol. The ECD spectra were obtained by weighing the Boltzmann distribution rate of each geometric conformation. SpecDis 1.61 [14] was used to sum up single CD spectra after a Boltzmann statistical weighting, and for Gauss curve generation and the comparison with experimental data.

### 3.6. Zebrafish Culture

Wild AB zebrafish were supported independently by the zebrafish experimental platform maintained at 28.5 °C water (pH 7.2–7.5; conductivity 500–550 μs/cm) under a 14 h light/10 h dark cycle. Healthy zebrafish, 4–8 months old, were paired to obtain zebrafish embryos and larvae. Zebrafish experimental operations were conducted according to the National Institutes of Health Guide for the Use and Care of Experimental Animals and were approved by the Animal Experimentation Ethics Committee of Chengdu University of TCM.

### 3.7. Gastrointestinal Motility of Zebrafish

Gastrointestinal motility assay was carried out with neostigmine (Appendix A) as positive control. In general, healthy 120 hpf (120 h postfertilization) zebrafish larvae were harvested and pre-treated with 10 ug/L Nile red. After dyeing for 16 h, the larvae were washed three times and exposed to 4, 8, 16, 32, and 64 μM of compounds **1**–**5**, with 0.1% DMSO as control. After 6 h treatment, zebrafish larvae were anaesthetized with tricaine, and immobilized on a slide with 1% CMC-Na. The fluorescence intensity of the zebrafish gastrointestinal tract was observed and photographed under fluorescence microscope. ImageJ software was used to obtain the IOD of fluorescence intensity.

## 4. Conclusions

Three new thiodiketopiperazines, along with two known analogues, had been isolated from a TCM-derived endophytic fungus *P. crustosum*. The absolute configurations of compounds 1–3 were determined by quantum chemical ECD calculations combined with Mosher ester analysis. Isopentene groups might be the key bioactive group for this kind of thiodiketopiperazine, to promote the gastrointestinal motility of zebrafish via acting on the cholinergic nervous system.

## Figures and Tables

**Figure 1 molecules-24-00299-f001:**
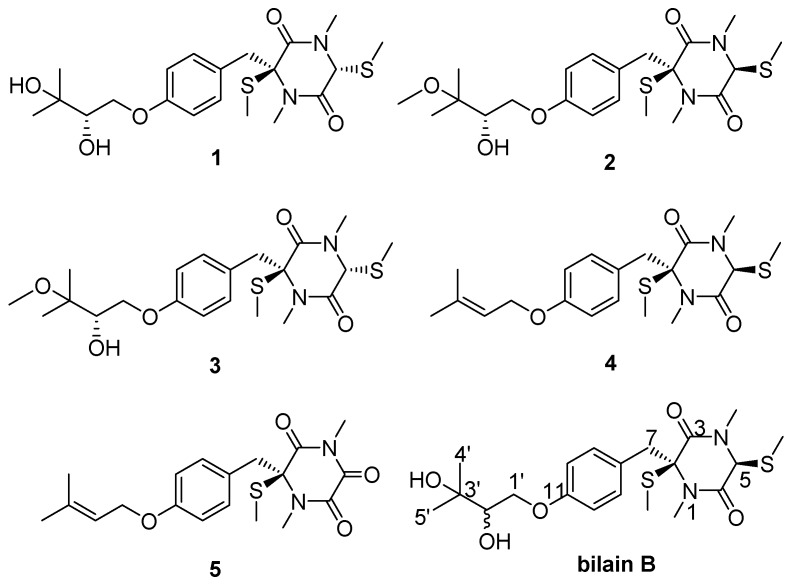
The structures of compounds 1–5 isolated from *P. crustosum* and bilain B.

**Figure 2 molecules-24-00299-f002:**
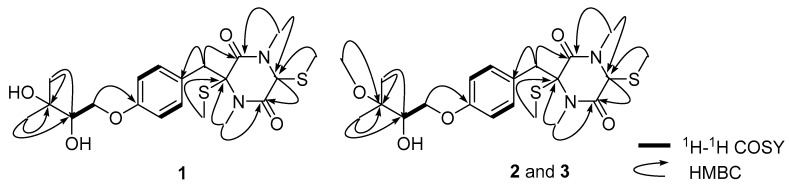
The ^1^H–^1^H COSY and key HMBC correlations of **1**–**3**.

**Figure 3 molecules-24-00299-f003:**
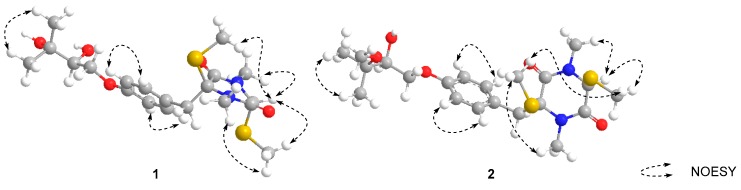
The key NOESY correlations of **1**–**2**.

**Figure 4 molecules-24-00299-f004:**
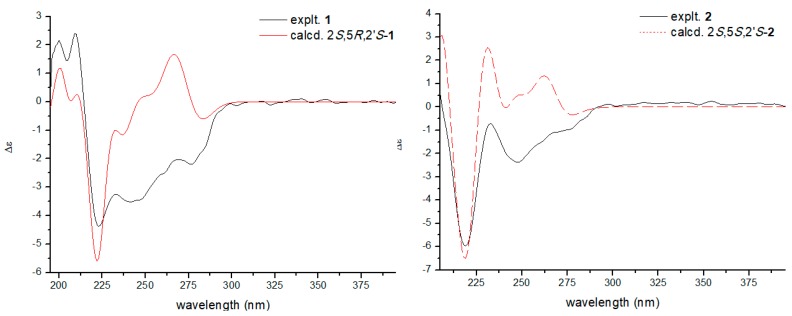
Calculated ECD spectra of **1** (4a) and **2** (4b) and their experimental curves.

**Figure 5 molecules-24-00299-f005:**
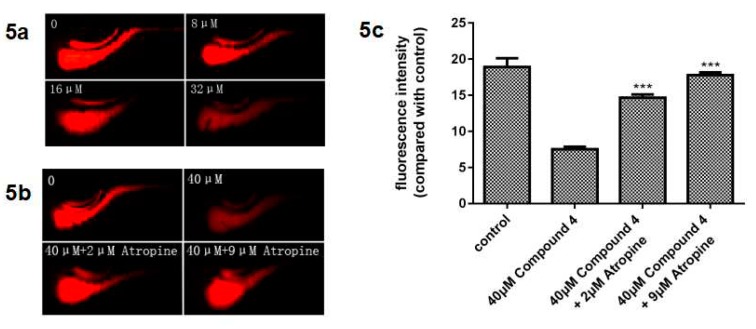
(**a**) Effect of compound **4** on the gastrointestinal motility of zebrafish. (**b**) and (**c**) Interfering effect of atropine on the gastrointestinal activation of compound **4**. *** indicates *p* < 0.001 when compared with the control.

**Figure 6 molecules-24-00299-f006:**
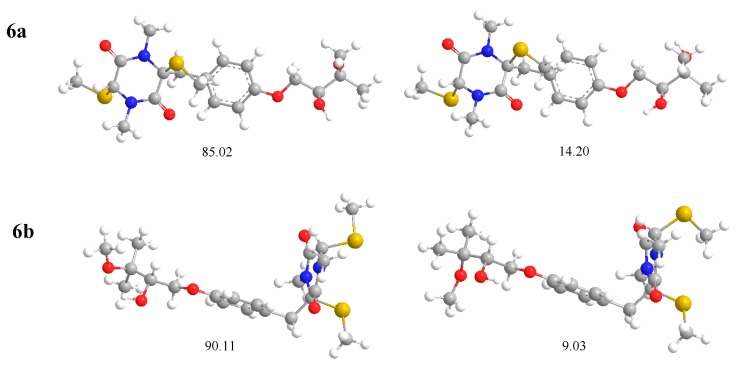
Most stable conformers of 2*S*,5*R*,2′*S*-**1** (**6a**) and 2*S*,5*S*,2′*S*-**2** (**6b**) calculated with DFT at the B3LYP/6-31G (d) level. Relative populations are in parentheses. Equilibrium populations calculated by the relative Gibbs free energies at B3LYP/6-31G (d) level in the gas phase, assuming Boltzmann statistics at T = 298.15 K and 1 atm.

**Table 1 molecules-24-00299-t001:** NMR data of **1**–**3**^a^.

Position	1	*δ* _C_	2	*δ* _C_	3	*δ* _C_
	*δ*_H_ (*J* in Hz)		*δ*_H_ (*J* in Hz)		*δ*_H_ (*J* in Hz)	
1	-	-	-	-	-	-
2	-	76.9	-	75.1	-	76.5
3	-	165.4	-	164.4	-	165.0
4	-	-	-	-	-	-
5	4.92, s	66.0	4.19, s	65.1	4.59, s	65.8
6	-	165.2	-	165.1	-	164.8
7	3.14, d (14.0) 3.58, d (14.0)	41.0	3.07, d (14.0) 3.53, d (14.0)	42.0	3.06, d (14.1) 3.65, d (14.1)	40.7
8	-	128.1	-	126.5	-	126.9
9,13	7.11, d (8.7)	132.5	6.97, d (8.7)	130.8	7.06, d (8.7)	131.6
10,12	6.85, d (8.7)	115.8	6.79, d (8.7)	114.7	6.82, d (8.7)	115.3
11	-	159.6	-	158.3	-	158.6
1′	4.23, dd (10.0, 3.0) 3.89, dd (10.0, 7.7)	70.7	4.07, dd (8.8, 3.2) 3.90, dd (14.1, 5.3)	69.1	4.08, dd (9.4, 3.1) 3.91, dd (9.4, 7.7)	69.2
2′	3.72, td (4.5, 2.2)	77.1	3.85-3.88, m	75.4	3.85, dd (7.6, 3.1)	75.3
3′	-	71.9	-	76.2	-	76.2
4′	1.23, s	26.7	1.23, s	21.0	1.22, s	21.0
5′	1.21, s	25.6	1.22, s	20.8	1.21, s	20.7
1-Me	3.20, s	30.6	3.25, s	30.3	3.04, s	33.5
2-SMe	1.95, s	12.6	2.16, s	13.7	1.95, s	12.7
4-Me	3.01, s	33.2	2.95, s	33.6	3.24, s	30.7
5-SMe	1.58, s	13.7	2.29	16.5	1.66, s	14.2
3’-OMe	-	-	3.26, s	49.4	3.25, s	49.4

a: **1** was measured in acetone-d_6_ and **2**, **3** were measured in chloroform-d (400 MHz for ^1^H and 100 MHz for ^13^C).

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
