# Peer review of "Thiodiketopiperazines Produced by Penicillium crustosum and Their Activities to Promote Gastrointestinal Motility"

_molecules, 2019, doi:10.3390/molecules24020299_

Round 1

Reviewer 1 Report

The authors have provided an account of the isolation of five thio-diketopiperazines (3 unknown and 2 "known") from Fungus Penicillium crustosum. The authors provide details of the full characterisation including sterochemical assignment of the three unknown compounds. The "known" compounds 4-5 were shown to promote the gastrointestinal motility of Zebrafish with compounds 1-3 appearing inactive which leads the author to conclude that the isopentene group might be the pharmacaphore. 

The strength of this paper is the full characterisation of identified thioketopiperizines numbered as 1-3. However, there are a number of issues with the manuscript as it stands which need to be addressed.  

Major corrections: 

The introduction is insufficient.

Compounds 4 and 5 are refered to as "known" in the abstract yet there is no mention of them in the introduction or a reference to their previous discovery at all. 

There is no introduction at all to the idea that these compounds can be used to promote gastrointestinal mobility. The first introduction is during the results and discussion and seems very out of place.

There is no information as to how the metabolites were isolated. More detail is required here or within the first paragraph of the results. A reader should not be expected to have to hunt to the methods where a rather complex proceedure is described.

The last line of the intoduction states "In this paper, the fermentation, isolation, structural elucidation and bioactivity of compounds 1-5 were described." Which is only true if you include the methods. Also again no information is really provided for 4-5. A reference is needed or this is not true as there is no evidence of any knowledge of the structures of 4-5.

Results

Line 109: The location of table 1 is poor. The only reference to this table is in the methods section so this would be better placed there or it should refered to in the main text. 

Experimental:

More detail is required to the column chromatography on silica gel where the extract was separated into 11 fractions by a gradient of petroleum ether-acetone which is not stated. Multiple fractions were collected and combined between 96:4 and 94:6? What was the full gradient used?

From the Sephadex LH-20 column fraction 2 was selected. Please provide a UV trace or more details as to why this was selected (i.e for e.g absorbance at 254nm).  

Minor gramatical/spelling corrections:

Line 88: should read "shares"

Line 112: Incorrect tense should read know

Line 120: Incorrect tense should read promote

Line 161: Subscript required in H2O

Line 170: Superscript required in 10-3 

Line 219: populations

Author Response

Thank you for your comments which are very helpful for revising and improving our paper. We have studied your comments carefully and have made revision which marked in red in the paper which we hope meet with approval. The responds to your comments are as following:

Major corrections:

The introduction is insufficient.

Compounds 4 and 5 are refered to as "known" in the abstract yet there is no mention of them in the introduction or a reference to their previous discovery at all.

There is no introduction at all to the idea that these compounds can be used to promote gastrointestinal mobility. The first introduction is during the results and discussion and seems very out of place.

There is no information as to how the metabolites were isolated. More detail is required here or within the first paragraph of the results. A reader should not be expected to have to hunt to the methods where a rather complex proceedure is described.

The last line of the intoduction states "In this paper, the fermentation, isolation, structural elucidation and bioactivity of compounds 1-5 were described." Which is only true if you include the methods. Also again no information is really provided for 4-5. A reference is needed or this is not true as there is no evidence of any knowledge of the structures of 4-5.

Response: Thank you for your professional comments. We rewrote the introduction, information about gastrointestinal mobility was described in lines 35-38 and the details about how these compounds were isolated were described in lines 41-43. The references about known compounds was mentioned in line 98.

Results

Line 109: The location of table 1 is poor. The only reference to this table is in the methods section so this would be better placed there or it should refered to in the main text.

Response: As your suggestion, table 1 was placed in the methods section.

Experimental:

More detail is required to the column chromatography on silica gel where the extract was separated into 11 fractions by a gradient of petroleum ether-acetone which is not stated. Multiple fractions were collected and combined between 96:4 and 94:6? What was the full gradient used?

Response: The detail of the full gradient was added in lines 155-156.

From the Sephadex LH-20 column fraction 2 was selected. Please provide a UV trace or more details as to why this was selected (i.e for e.g absorbance at 254nm).

Response: As we can see from the HPLC profile of fraction 2, these diketopiperazines have absorbance at 210nm (black line) and 254nm (red line).

gradient of this HPLC profile is as following:

Time

 Methanol (ml)

Water (ml)

0

0.5

0.5

30

1

0

35

1

0

40

0.5

0.5

Minor gramatical/spelling corrections:

Line 88: should read "shares"

Response: We are sorry for our incorrect writing and this was revised in line 85.

Line 112: Incorrect tense should read know

Response: We are sorry for our incorrect writing and this was revised in line 111.

Line 120: Incorrect tense should read promote

Response: We are sorry for our incorrect writing and this was revised in line 119.

Line 161: Subscript required in H2O

Response: We are sorry for our incorrect writing and this was revised in line 151.

Line 170: Superscript required in 10-3

Response: We are sorry for our incorrect writing and this was revised in line 164.

Line 219: populations

Response: We are sorry for our incorrect writing and this was revised in line 206.

Finally, We appreciate your warm work earnestly, and hope that the correction will meet with approval. Once again, thank you very much for your comments and suggestion!

Reviewer 2 Report

Two features of the title are a cause of concern. Description of P. crustosum as 'endophytic' leads one to expect that this is some very peculiar form of the fungus. P. crustosum is a widespread saprophyte occurring and living in soil and on any moist nutritious substrate. In these circumstances it is not endophytic. A search engine shows a few literature items in recent years that claim endophytic status for P. crustosum in a few instances. Frankly, I am doubtful that these have been rigorously demonstrated for the comfortable, intimate, persistent and mutually-integrated association normally characteristic of an endophyte.  No such demonstration of endophytic status has been presented for the P. crustosum isolate of the present study and I do not think the descriptor should be used in the present manuscript.

The second cause for concern is Isaria cicadae. It could imply that the endophyte had that relationship with the fruiting body of another fungus that, by definition, had its own parasitic relationship with a dead Cicada. Maybe authors did not intend that. I assume there was a desire to highlight the origin from that TCM-derived (line 242) endophyte. Consequently I do not think that name should be in the title.  It is OK in Methods to say where the Penicillium came from, out of general interest,  since no special medical attribute is being assigned to that fungus-fungus association.

I see no problem with the natural products chemistry, but state that I am not expert in some of the modern methodology.

A few typograhical observations:

L 22 suggest delete endophyte

   42  fraction

    92 diketopiperazine

   112 known

   120 might promoted...language?

   149 seed medium PDA....presumably PD  liquid

   180 compounds

   227 larvae.....maybe journal style favours continuity of the plural in latin.

Since fungal diketopiperazines are conventionally biosynthesised by condensation of two amino acids, authors might like briefly to suggest the sources for the present compounds.

Author Response

Thank you for your comments which are very helpful for revising and improving our paper. We have studied your comments carefully and have made revision which marked in red in the paper which we hope meet with approval. The responds to your comments are as following:

Two features of the title are a cause of concern. Description of P. crustosum as 'endophytic' leads one to expect that this is some very peculiar form of the fungus. P. crustosum is a widespread saprophyte occurring and living in soil and on any moist nutritious substrate. In these circumstances it is not endophytic. A search engine shows a few literature items in recent years that claim endophytic status for P. crustosum in a few instances. Frankly, I am doubtful that these have been rigorously demonstrated for the comfortable, intimate, persistent and mutually-integrated association normally characteristic of an endophyte. No such demonstration of endophytic status has been presented for the P. crustosum isolate of the present study and I do not think the descriptor should be used in the present manuscript.

The second cause for concern is Isaria cicadae. It could imply that the endophyte had that relationship with the fruiting body of another fungus that, by definition, had its own parasitic relationship with a dead Cicada. Maybe authors did not intend that. I assume there was a desire to highlight the origin from that TCM-derived (line 242) endophyte. Consequently I do not think that name should be in the title. It is OK in Methods to say where the Penicillium came from, out of general interest, since no special medical attribute is being assigned to that fungus-fungus association.

Response: Thank you for your professional comments. It is really ture as you suggested that P. crustosum is a widespread saprophyte, we use the words Isaria cicadae-colonizing fungus” to describe this strain. Due to the fact that there is no special medical attribute is being assigned to that fungus-fungus association, Isaria cicadae was eliminated from the title.

I see no problem with the natural products chemistry, but state that I am not expert in some of the modern methodology.

The responds to typing mistakes are as following:

A few typograhical observations:

L 22 suggest delete endophyte

Response: As your suggestion, this was revised and endophyte was deleted.

42 fraction

Response: We are sorry for our incorrect writing and this was revised in line 42.

 92 diketopiperazine

Response: We are sorry for our incorrect writing and this was revised in line 93.

112 known

Response: We are sorry for our incorrect writing and this was revised in line 111.

120 might promoted...language?

Response: This was revised in line 119.

149 seed medium PDA....presumably PD liquid

Response: This was revised in line 148.

180 compounds

Response: This was revised in line 180.

227 larvae.....maybe journal style favours continuity of the plural in latin.

Response: It is really ture as your suggestion and this was revised in line 230.

Since fungal diketopiperazines are conventionally biosynthesised by condensation of two amino acids, authors might like briefly to suggest the sources for the present compounds.

Response: It is really ture as you suggested that condensation of two or three amino acid precursors is involvedin the DKP biosynthetic pathway with the later also being known as cyclic dipeptides when derived from condensation of two amino acids.

It was reported that secondary metabolites from fungi that incorporate more than one amino acid are typically synthesized via non-ribosomal peptide synthetases (NRPSs) [1]. A number of DKP derivatives feature either thiomethyl groups or disulphide bridges [2], and those with an internal disulphide bridge are known as epipolythiodioxopiperazines (ETPs) which are typically toxic metabolites from fungi.

References:

1.Mootz, H.D., Marahiel, M.A., Biosynthetic systems for nonribosomal peptide antibiotic assembly. Curr. Opin. Chem. Biol. 1997, 1, 543-551.

2.Gardiner, D.M., Waring, P., Howlett, B., The epipolythiodioxopiperazine (ETP) class of fungal toxins: distribution, mode of action, functions and biosynthesis. J. Microbiol. 2005, 151, 10211032.

Reviewer 3 Report

This manuscript describes that structure elucidation of thio-diketopiperazines produced by Penicillium crustosum and their activity to promote gastrointestinal motility. The structural determination containing the absolute configurations of new compounds 1-3 was achieved by use of 2D NMR, modified Mosher’s method, and calculated ECD analysis. It is interesting that compounds 4 and 5 exhibit the activity to promote the gastrointestinal motility of zebrafish. Therefore, this manuscript is recommended to be published in Molecules after the following points are considered.

1) Line 54: “(2H, m, H-10 and H-12)” was revised to “(2H, d, J = 8.7 Hz, H-10 and H-12)”

2) Line 54: “(2H, m, H-9 and H-13)” was revised to “(2H, d, J = 8.7 Hz, H-9 and H-13)”

3) Line 59: “165.4 (C-1) and 165.2 (C-4)” was revised to “165.4 (C-3) and 165.2 (C-6)”

4) Line 62: “33.2 (6-Me), 30.6 (3-Me)” was revised to “33.2 (4-Me), 30.6 (1-Me)”

5) Line 72: The NOESY correlation between H-5 and 2-SMe was not observed according to Figure 3.

6) Line 81: The NOESY correlation between 2-SMe and 5-SMe was not observed according to Figure 3.

7) NOESY correlations and ECD spectrum of compound 3 should be described.

8) The authors should represent the activity of a positive control to help readers assess whether the activities of compounds 4 and 5 are strong or not.

9) Lines 166, 170, and 174: The solvent used the UV spectral analysis should be described.

Author Response

Thank you for your comments which are very helpful for improving our paper. We have studied your comments carefully and have made revision which marked in red in the paper which we hope meet with approval. The responds to your comments are as following:

1. Line 54: “(2H, m, H-10 and H-12)” was revised to “(2H, d, J = 8.7 Hz, H-10 and H-12)”

Response: This was revised in line 56.

2. Line 54: “(2H, m, H-9 and H-13)” was revised to “(2H, d, J = 8.7 Hz, H-9 and H-13)”

Response: This was revised in line 56.

3. Line 59: “165.4 (C-1) and 165.2 (C-4)” was revised to “165.4 (C-3) and 165.2 (C-6)”

Response: This was revised in line 62

4. Line 62: “33.2 (6-Me), 30.6 (3-Me)” was revised to “33.2 (4-Me), 30.6 (1-Me)”

Response: This was revised in line 65

5. Line 72: The NOESY correlation between H-5 and 2-SMe was not observed according to Figure 3.

Response: Figure 3 was redrew, The NOESY correlation of H-5/2-SMe was added.

6. Line 81: The NOESY correlation between 2-SMe and 5-SMe was not observed according to Figure 3.

Response: Figure 3 was redrew, The NOESY correlation of 2-SMe/5-SMe was added.

7. NOESY correlations and ECD spectrum of compound 3 should be described.

Response: The NOESY correlations and ECD spectrum of compound 3 were described in lines 91-93.

8. The authors should represent the activity of a positive control to help readers assess whether the activities of compounds 4 and 5 are strong or not.

Response: Gastrointestinal motility assay was carried out with neostigmine as positive control. Neostigmine promoted the Nile red excretion at doses > 2 μM. As your suggestion, this information was described in lines 103 and 237.

9. Lines 166, 170, and 174: The solvent used the UV spectral analysis should be described.

Response: The solvent used the UV spectral analysis was add in lines 166, 170 and 174.

Once again, thank you very much for your comments and suggestion.

Round 2

Reviewer 1 Report

My thanks to the authors for addressing all of my previous concerns.